# Cardiac Autonomic Function and Functional Capacity in Post-COVID-19 Individuals with Systemic Arterial Hypertension

**DOI:** 10.3390/jpm13091391

**Published:** 2023-09-18

**Authors:** Edelvita Fernanda Duarte Cunha, Matheus Sobral Silveira, Juliana Cristina Milan-Mattos, Heitor Fernandes Silveira Cavalini, Ádrya Aryelle Ferreira, Joice de Souza Batista, Lara Cazé Uzumaki, João Paulo Coelho Guimarães, Pedro Igor Lustosa Roriz, Fabianne Maisa de Novaes Assis Dantas, Arto J. Hautala, Raphael Martins de Abreu, Aparecida Maria Catai, Paulo Adriano Schwingel, Victor Ribeiro Neves

**Affiliations:** 1Programa de Pós-Graduação em Reabilitação e Desempenho Funcional (PPGRDF), Universidade de Pernambuco (UPE), Petrolina 56328-900, PE, Brazil; edelvita.fernanda@upe.br (E.F.D.C.); matheus.sobrals@upe.br (M.S.S.); heitorcavalini@gmail.com (H.F.S.C.); adrya.ferreira@upe.br (Á.A.F.); pedro.roriz@upe.br (P.I.L.R.); 2Grupo de Estudos e Pesquisas em Fisioterapia Cardiorrespiratória (GEFIC), Universidade de Pernambuco (UPE), Petrolina 56328-900, PE, Brazil; fisiojoice.s@gmail.com (J.d.S.B.); lara.uzumaki@upe.br (L.C.U.); joao.pcguimaraes@upe.br (J.P.C.G.); fabianne.dantas@upe.br (F.M.d.N.A.D.); mcatai@ufscar.br (A.M.C.); 3Laboratório de Fisioterapia Cardiopulmonar (LAFIC), Universidade de Pernambuco (UPE), Petrolina 56328-900, PE, Brazil; 4Laboratório de Pesquisas em Desempenho Humano (LAPEDH), Universidade de Pernambuco (UPE), Petrolina 56328-900, PE, Brazil; 5Postgraduate Program in Physical Therapy (PPGFT), Federal University of São Carlos (UFSCar), São Carlos 13565-905, SP, Brazil; 6Faculty of Sport and Health Sciences, University of Jyväskylä, P. O. Box 35, FI-40014 Jyväskylä, Finland; arto.j.hautala@jyu.fi; 7Department of Physiotherapy, LUNEX University—International University of Health, Exercise & Sports SA, 4671 Differdange, Luxembourg; raphael.martinsdeabreu@lunex-university.net; 8LUNEX ASBL Luxembourg Health & Sport Sciences Research Institute, 4671 Differdange, Luxembourg

**Keywords:** hypertension, SARS-CoV-2, post-COVID conditions, physical functional performance, autonomic nervous system diseases, heart rate control

## Abstract

Individuals diagnosed with systemic arterial hypertension (SAH) are considered risk groups for COVID-19 severity. This study assessed differences in cardiac autonomic function (CAF) and functional capacity (FC) in SAH individuals without COVID-19 infection compared to SAH individuals post-COVID-19. Participants comprised 40 SAH individuals aged 31 to 80 years old, grouped as SAH with COVID-19 (G1; n = 21) and SAH without COVID-19 (G2; n = 19). CAF was assessed via heart rate variability (HRV), measuring R–R intervals during a 10-min supine period. Four HRV indices were analyzed through symbolic analysis: 0V%, 1V%, 2LV%, and 2UV%. FC assessment was performed by a 6-min walk test (6MWT). G1 and G2 showed no significant differences in terms of age, anthropometric parameters, clinical presentation, and medication use. G2 exhibited superior 6MWT performance, covering more distance (522 ± 78 vs. 465 ± 59 m, *p* < 0.05). Specifically, G2 demonstrated a moderate positive correlation between 6MWT and the 2LV% index (r = 0.58; *p* < 0.05). Shorter walking distances were observed during 6MWT in SAH individuals post-COVID-19. However, the study did not find impaired cardiac autonomic function in SAH individuals post-COVID-19 compared to those without. This suggests that while COVID-19 impacted FC, CAF remained relatively stable in this population.

## 1. Introduction

Systemic arterial hypertension (SAH) is considered one of the most prevalent risk factors for the development of coronavirus disease 2019 (COVID-19). Hypertensive individuals are classified as high-risk groups due to their strong association with disease severity and mortality [1,2,3,4]. Studies report that this association is due to the high affinity of SARS-CoV-2 for the angiotensin-converting enzyme 2 (ACE2) receptor, an enzyme highly expressed in these individuals, rendering them more susceptible to the disease. This promotes endothelial cell dysfunction, elevated blood pressure, and other cardiovascular abnormalities [5,6].

The inflammatory process triggered during the disease disrupts the cholinergic anti-inflammatory pathway. This pathway controls cytokine release through afferent vagal stimulation, as cytokine release is controlled by the parasympathetic branch of the autonomic nervous system (ANS). However, hypertensive individuals often present with a pre-existing sympathovagal imbalance, which might be interpreted as an adaptive condition to maintain homeostasis [7,8,9]. Given this imbalance, an adequate anti-inflammatory response to SARS-CoV-2 in these individuals appears unlikely, as autonomic imbalance could be associated with an inflammatory cytokine storm during the disease progression [1,10,11].

In addition to the imbalance in cardiac autonomic function (CAF), patients after the acute phase of SARS-CoV-2 infection may experience persistent and debilitating symptoms such as chronic fatigue, dyspnea, myalgia, arrhythmias, chest pain, orthostatic intolerance, and exercise intolerance. These symptoms lead to reduced functional capacity and worsened quality of life [11,12,13].

As reported in the literature, individuals with cardiovascular diseases (CVDs) and associated risk factors such as SAH experience a decline in functional capacity (FC) due to exercise intolerance, leading to a significant impact on their quality of life. This results from their physical inability to perform daily and occupational activities. When combined with the functional limitations imposed by COVID-19, even in mild cases, these individuals might experience further deterioration [13,14]. In this study, the aim was to assess differences in CAF and FC in SAH individuals without COVID-19 infection compared to SAH individuals post-COVID-19.

## 2. Materials and Methods

### 2.1. Study Design

The observational study was conducted from July 2021 to September 2022 at the University of Pernambuco (UPE) in Petrolina, PE, Brazil. The research was approved by the Research Ethics Committee of the Centro Integrado de Saúde Amaury de Medeiros (protocol 4848824 on 29 June 2021), and all participants provided voluntary consent prior to participation. Eligible individuals were recruited in Petrolina, facilitated by the Municipal Health Department (SESAU), based on a list of those diagnosed with systemic arterial hypertension (SAH) and infected with SARS-CoV-2. The study was promoted through various media channels and the assessments were carried out at the UPE Cardiopulmonary Physiotherapy Laboratory, with assessors being blinded to participant groups.

### 2.2. Participants

Individuals of both genders, aged between 31 and 80 years, diagnosed with SAH for at least one year, and on continuous antihypertensive medication for at least three months were evaluated in this study. These individuals were either infected or not with COVID-19. Participants were divided into two groups as follows:Group 1 (G1): SAH patients infected with SARS-CoV-2, with confirmed diagnosis through the reverse transcription polymerase chain reaction (RT-PCR), within six months of test confirmation, asymptomatic or experiencing mild symptoms of COVID-19 and fully recovered of infection.Group 2 (G2): Individuals with SAH without a clinical diagnosis of COVID-19, and with a negative RT-PCR test confirming the absence of the virus.

Individuals were not included if they experienced moderate to severe COVID-19 symptoms, had pre-existing diagnoses or neurological sequelae of diseases, had chronic obstructive pulmonary disease (COPD), had blood pressure exceeding 180/100 mmHg, were pregnant, had physical or mental limitations at the time of assessment, or had significant clinical conditions that contraindicated the tests. Volunteers who were on medications affecting the studied variables’ responses and who were unable to complete the tests and/or assessment protocols were also excluded. Participants with poor-quality electrocardiogram (ECG) signals (interference, signal instability) were excluded from the research. Furthermore, participants classified as having good, excellent, or superior cardiorespiratory fitness (CRF) during the six-minute walk test (6MWT) were also excluded (Figure 1).

### 2.3. Assessments

#### 2.3.1. Anthropometric and Hemodynamic Parameters Measurements

Participants’ total body mass was measured in kilograms (kg), and height was measured in centimeters (cm) using a properly calibrated anthropometric scale (PL-200, Filizola S.A., São Paulo, SP, Brazil), adhering to the standards set by NBR ISO/IEC 17025:2005. Body mass index (BMI) was calculated by dividing body mass (kg) by the square of height (m^2^). For the measurement of systolic blood pressure (SBP) and diastolic blood pressure (DBP), an automatic, noninvasive, calibrated, and validated arterial blood pressure monitor, HEM-7200 (Omron Healthcare Inc., Lake Forest, IL, USA), was employed. The measurement procedures followed the guidelines outlined in the Brazilian Guidelines of Hypertension [15].

#### 2.3.2. Evaluation of Cardiac Autonomic Control System

The CAF was evaluated by heart rate variability (HRV) assessment and was conducted in the supine position for 10 min. The R–R interval (iRR) recordings from the ECG were acquired using a twelve-lead electrocardiograph Wincardio (MICROMED BIOTECNOLOGIA S.A., Brasília, DF, Brazil). A sequence of 256 iRR with the highest stability was chosen for each subject to calculate the mean, variance of iRR, and HRV indices in the frequency domain. All assessment methodologies of HRV were analyzed in specific routines, developed by Prof. Dr. Alberto Porta of the University of Milan, Italy [16].

Frequency domain HRV analysis was executed using an autoregressive model applied to the previously selected iRR sequence data [17]. Two spectral components were considered: low frequency (LF—0.04 to 0.15 Hz), representing mixed (sympathetic and parasympathetic) modulation, and high frequency (HF—0.15 to 0.50 Hz), representing parasympathetic cardiac modulation [18]. The spectral components were expressed in normalized units (LFnu and HFnu) and absolute units (LFabs and HFabs). Respiratory frequency (RF) was evaluated during the protocol to ensure the presence of respiratory activity within its corresponding bandwidth for spectral analysis. All the volunteers presented a respiratory rate in the HF range.

Symbolic analysis (SA) grouped all patterns into four families: (a) patterns without variation (0V); (b) patterns with one variation (1V: two consecutive symbols are the same, and one symbol is different); (c) patterns with two similar variations (2LV); (d) patterns with two different variations (2UV: three symbols forming a peak or valley). The occurrence rate of each pattern is denoted as 0V%, 1V%, 2LV%, and 2UV%. Here, 0V% and 2UV% are considered markers of sympathetic and vagal modulation, respectively [16,19].

The complexity analysis was executed following the methodology proposed by Porta et al. [20]. The analysis encompassed the evaluation of Shannon entropy (SE) and conditional entropy (CE), in addition to the assessment of the complexity index (CI) and normalized complexity index (NCI).

#### 2.3.3. Evaluation of FC

The 6MWT was employed to assess FC by the American Thoracic Society (ATS) guidelines [21]. A 30-m-long corridor devoid of obstacles was utilized for the test, marked every 3 m with cones at the ends. The participant was familiarized with the path before the walk commenced. After six minutes, the distance covered by the individual in meters (6-min walk distance—6MWD) was recorded [21,22]. Heart rate and oxygen saturation (SpO2) were recorded with the finger pulse oximeter OXIOLCM (Beijing Choice Electronic Technology Co., Ltd., Beijing, China). At the beginning and end of the experiment, the blood pressure of all volunteers was measured by the auscultatory method. The rate of perceived exertion (RPE) at the end of the 6MWT was recorded with the modified 0–10 Borg Scale [23]. The reference equation used in this research for the prediction of the total distance to be walked during the 6MWT was developed by Iwama et al. [24]:6MWD = 622.461 − (1.846 × Age years) + (61.503 × Gender males = 1; females = 0)

According to Dourado et al. [25], fitness levels were classified using CRF and the 6MWD covered in meters during the 6MWT, stratified by age groups. The levels were categorized as very low, low, fair, good, excellent, and superior.

### 2.4. Statistical Analysis

The data were processed and analyzed using the SigmaPlot software (Systat Software Inc., San Jose, CA, USA, Release 11.0, 2011). The normality of data distribution was assessed using the Shapiro–Wilk test. Categorical variables were compared by Pearson’s chi-squared test (*Χ*^2^) or Fisher’s exact test and expressed as percentages and frequency. For continuous variables, the independent samples *t*-test was applied to parametric variables, and the Mann–Whitney U test was used for non-parametric data. To explore the relationship between functional testing and HRV indices, Spearman’s rank correlation coefficient was utilized for non-parametric variables, while Pearson’s product–moment correlation coefficient was used for parametric variables. All statistical methods were two-tailed, with exact *p* values, and statistical significance was defined as *p* < 0.05 [26]. The strength of correlations was categorized as follows: 0.00–0.25, very low; 0.26–0.49, low; 0.50–0.69, moderate; 0.70–0.89, high; 0.90–1.00, very high [26]. Additionally, Cohen’s d-effect size (ES) was employed to estimate the magnitude of differences between groups [27]. The ES was categorized as follows: 0.2–0.3, small; 0.5–0.8, medium; >0.8, large [27].

## 3. Results

### 3.1. Sample Characteristics

Out of the initial pool of 113 volunteers who fulfilled the eligibility criteria, a refined group of 40 individuals was subjected to evaluation, following the application of exclusion criteria. It is important to highlight that all study participants were under continuous medication for controlling SAH. The comprehensive overview of the study’s participant profile is depicted in Table 1. Notably, no statistically significant differences emerged between the two groups in terms of crucial attributes, including age, gender, height, total body mass, and BMI.

Regarding the COVID-19 symptoms reported by volunteers in G1 during the disease, the majority experienced cough (57%), headache (48%), fever (48%), anosmia (43%), runny nose (33%), shortness of breath (19%), fatigue (14%), and myalgia (10%).

### 3.2. Cardiac Autonomic Function

The HRV indices of the studied groups are presented in Table 2. No statistical differences were observed in the linear and non-linear HRV indices between the groups with very low ES (*d* < 0.5). The groups exhibited similarity in RF, with G1 presenting an RF of 18 ± 4 breaths per minute (bpm) and G2 exhibiting an RF of 17 ± 4 bpm (*p* > 0.05). It is worth noting that in the spectral analysis data, both groups displayed a predominance of high frequency (HF), with no statistical difference between them. Regarding the symbolic analysis, both groups demonstrated a predominance of the 1V% pattern, which represents mixed (sympathetic and parasympathetic) modulation, with no statistical difference between the groups.

### 3.3. Functional Capacity

Of the forty study participants, eight (20%) did not complete the 6MWT protocol (three in the G1 and five from G2). Within G1, three (14%) participants were classified as having a level of cardiorespiratory fitness (CRF) equal to or higher than good and were subsequently excluded from all analyses. Consequently, for the variables obtained through the 6MWT, the groups were composed as follows: G1 with 18 post-COVID-19 individuals with SAH (10 males; 51 ± 11 years; 88.1 ± 23.5 kg; 1.64 ± 0.1 m; BMI: 31.03 ± 10.4 kg/m^2^); G2 with 14 individuals with SAH (9 males; 56 ± 10 years; 81.9 ± 14.5 kg; 1.70 ± 0.1 m; BMI: 29.3 ± 2.8 kg/m^2^). These variables did not exhibit statistically significant differences in the comparison conducted using the independent-samples *t*-test.

The distance covered during the 6MWT displayed a statistically significant difference between G2 and G1 (as shown in Table 3), with G2 covering a significantly greater distance (*p* = 0.0241). The mean difference indicated that G2 walked 58 m more than G1, illustrating a large ES of *d* > 0.8. Conversely, there were no statistically significant differences observed between the groups in relation to other variables under analysis, including heart rate frequency, systolic blood pressure, diastolic blood pressure, blood oxygen level, Borg rating of perceived exertion during the 6MWT, and the percentage difference between predicted and covered distances on the 6MWT. Importantly, both G1 and G2 fell short of achieving the full percentage of predicted distance, with G1 reaching 85 ± 12% and G2 achieving 93 ± 10% (*p* = 0.0489).

The fitness levels classified eight (44%) post-COVID-19 individuals with SAH (G1) as very low CRF, seven (39%) as low CRF, and three (17%) as fair CRF. In contrast, only one (7%) individual with SAH (G2) presented very low CRF, while seven (50%) had low CRF and six (43%) had fair CRF.

Figure 2 illustrates the correlation between the distance covered in the 6MWT and the 2UV% index in the studied groups. A positive and moderate correlation was observed between these variables solely in G2, indicating that greater parasympathetic modulation was associated with increased distance covered (*r* = 0.58; *p* < 0.05). No significant correlations were observed for other variables in both groups.

## 4. Discussion

The current study aimed to assess differences in cardiac autonomic modulation and functional capacity in individuals with systemic arterial hypertension without COVID-19 infection compared to post-COVID-19 individuals. The main finding of this study was that post-COVID-19 individuals with SAH exhibited a reduced distance covered on the 6MWT compared to SAH individuals without clinical diagnosis of COVID-19. Thus, SAH patients infected with SARS-CoV-2, even with mild symptoms, experienced a decline in FC. Additionally, the greater physical capacity observed in SAH individuals without COVID-19 showed a positive correlation with higher parasympathetic modulation at rest. Ultimately, there were no significant differences in linear and non-linear HRV indices between the groups.

The 6MWT has been widely utilized to evaluate the functional capacity of individuals with various conditions, including cardiovascular, respiratory, and metabolic diseases, as well as those who have recovered from COVID-19 [28,29]. Wong et al. [30], studying post-COVID-19 patients, found an association between the severity of the disease and the distance covered on 6MWT, where individuals with moderate to severe symptoms covered less distance compared to those with mild symptoms. The participants in that study (31 men and 32 women, 42 ± 11 years old) were hospitalized and did not report hypoxemia, with only a few risk factors such as hypertension (13%) and diabetes mellitus (5%). This group covered a mean distance of 491 (±72) m in the 6MWT, achieving a distance predicted of 83%. In our study, none of the post-COVID-19 participants were hospitalized; all had mild symptoms and covered a mean distance of 465 (±60) m, 27 m less in mean than the study mentioned above. This discrepancy may be attributed to the fact that the study performed by Wong et al. [30] was conducted on a younger population with fewer associated risk factors. Furthermore, all participants in the present study had a diagnosis of SAH, which is considered a significant risk factor associated with COVID-19 complications.

In a retrospective study, Huang et al. [31] investigated the influence of COVID-19 on physical capacity during the early convalescent phase and 30 days post-hospital discharge. They evaluated two groups based on clinical severity, severe (n = 17) and non-severe (n = 40), who underwent the 6MWT. According to their results, the non-severe group covered 574 (±38) m, corresponding to a distance predicted of 96%. Another study assessed the impact of COVID-19 on functional capacity in 87 patients with mild COVID-19 symptoms after 60 days of symptom onset and hospital discharge. The 6MWT distance for this group was 538 (±57) m [30]. In the present study, the covered distance on 6MWT was lower than in the studies mentioned above. The participants from previous studies presented mild COVID-19 symptoms but were hospitalized, which did not occur in the present study. Additionally, the discrepancy in 6MWT values could be attributed to the profile of the populations in the cited studies. This study evaluated only SAH individuals who had mild COVID-19 symptoms and were not hospitalized. Considering these results, even mild COVID-19 can lead to a decline in functional capacity, especially in SAH individuals. Although the underlying pathophysiological mechanisms remain unclear, COVID-19 may negatively influence physical capacity, even in individuals with mild clinical presentation.

Regarding risk factors, several studies have described SAH as one of the most common factors associated with disease severity. However, the pathophysiological mechanism of hypertension and its relationship with COVID-19, as well as its impact on physical capacity, remain unclear [2]. In the present study, participants with SAH without COVID-19 covered nearly 60 m more on the 6MWT than post-COVID-19 individuals with SAH. A systematic review demonstrated that an increase of 37 m in 6MWT distance is clinically significant [32]. Therefore, we infer that even mild COVID-19 in SAH individuals may contribute to a decline in FC assessed by the 6MWT. Further studies with larger sample sizes are needed to confirm these findings.

In addition to the observed higher physical capacity in G2, a positive correlation was found between the distance covered on the 6MWT and the 2LV% HRV index. According to Porta et al. [20], this index corresponds to vagal modulation in the heart. Regular physical activity has been shown to positively affect the vagal system [33]. Thus, non-infected SAH individuals showed better physical capacity as assessed by the 6MWT due to greater vagal modulation.

Asarcikli et al. [34] evaluated both healthy and recovered COVID-19 individuals, performing 24-h Holter monitoring for HRV analysis. They observed significantly higher rMSSD, PNN50, and HF indices in the post-COVID-19 group. These authors justified that even 12 weeks after infection, increased parasympathetic tone can persist. Furthermore, they suggested that prolonged parasympathetic activity might be associated with chronic symptoms such as dyspnea, fatigue, and arthralgia, which affect physical capacity and quality of life. Another study compared 63 patients with mild to moderate COVID-19 symptoms to 43 healthy individuals matched for age and sex [35]. Increased parasympathetic modulation was observed in COVID-19 patients, independent of important factors like age, sex, and comorbidities. Authors reported alterations in HRV indices such as rMSSD and SDNN, as well as frequency domain indices (LF and HF) [35]. Additionally, 33% of the patients in the studied group had hypertension.

On the other hand, Marques et al. [36] studied 155 patients with long COVID and 94 healthy volunteers. They observed an overall reduction in HRV, with an increase in sympathetic modulation and a decrease in parasympathetic modulation in the COVID-19 group. The authors concluded that COVID-19 led to an increase in sympathetic influence at rest and a decrease in vagal modulation compared to the non-infected group, and that reduced vagal activity might be related to mediation of the inflammatory process.

In the present study, no differences in cardiac autonomic control were observed between SAH patients with or without COVID-19. While hypertension and reduced physical capacity can influence reduced HRV with increased sympathetic and decreased vagal modulation [8,37], it is still uncertain how COVID-19 affects hypertensive individuals. We hypothesized that COVID-19 might further reduce HRV due to the inflammatory process, even in mild cases of the disease [34,35]. These factors, even when considering that the non-COVID-19 group had better physical capacity, may have contributed to the absence of differences in HRV results.

Regarding cardiac autonomic control during the post-COVID-19 period, the literature remains contradictory. This could be due to the heterogeneity of the samples studied to date, varying times of analysis post-infection, as well as different methods of HRV analysis. Although this study considered linear and non-linear HRV approaches to assess CAF, the impact of the disease on the autonomic nervous system may be limited when univariate HRV analysis is applied [38]. The nature of COVID-19 is not fully understood and its impact on the cardiac neural regulation may be driven by changes in other subsystems, such as the respiratory and vascular systems. A previous study reported that changes in CAF, when comparing patients with serious and mild COVID-19, were only observed when bivariate analysis of heart rate and systolic blood pressure variabilities were considered [38]. Additionally, it is known that HRV fluctuates spontaneously according to respiratory cycles, which may be determined by physiological, mechanical, and neural mechanisms, such as changes in intrathoracic pressure, the central influence of respiratory centers modulating vagal motoneuron activity, as well as changes in stroke volume during respiratory phases [39]. Therefore, it seems reasonable that changes in pulmonary diffusion as well as lung capacities caused by the disease may affect the cardiac regulation driven by respiration, called cardiorespiratory coupling [40,41]. Thus, multivariate approaches considering blood pressure oscillations as well as cardiorespiratory coupling analysis would be useful in the future to elucidate the impact of COVID-19 on the autonomic nervous system.

The main limitations of this study included difficulties in participant recruitment due to pandemic-related factors, which resulted in a reduced sample size and potential bias in HRV data. The variation in the time post-SARS-CoV-2 infection for the assessment was not consistent among all participants (up to 6 months post-infection), which is a notable limitation. Also, our sample primarily comprised asymptomatic patients or individuals with mild symptoms during their infection. Consequently, they did not report lasting symptoms typically associated with post-COVID syndrome, especially those who were either hospitalized or received outpatient treatment [42,43]. Furthermore, the lack of gender matching may have also had an influence. Finally, the absence of a healthy control group without COVID-19 and hypertension limits our ability to make comprehensive comparisons.

On the other hand, the results of the study can have clinical implications for various healthcare professionals in guiding rehabilitation programs for individuals and improving their physical capacity. Additionally, this study provides insight into hypertensive individuals, a population with a high prevalence of complications and mortality associated with COVID-19. The results contribute to the scientific knowledge in this field and emphasize the need for more research to better understand the impact of COVID-19 on individuals with pre-existing conditions.

## 5. Conclusions

This study demonstrated the negative impact of post-COVID-19 on the functional capacity of medicated hypertensive individuals who experienced a mild clinical presentation of the disease. However, the study did not find impaired cardiac autonomic function in systemic arterial hypertension individuals post-COVID-19 compared to those without COVID-19.

## Figures and Tables

**Figure 1 jpm-13-01391-f001:**
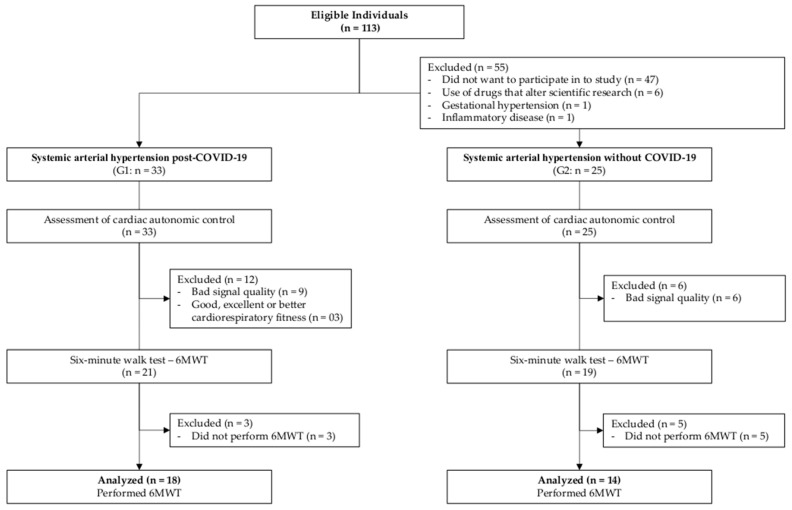
Flowchart of the study.

**Figure 2 jpm-13-01391-f002:**
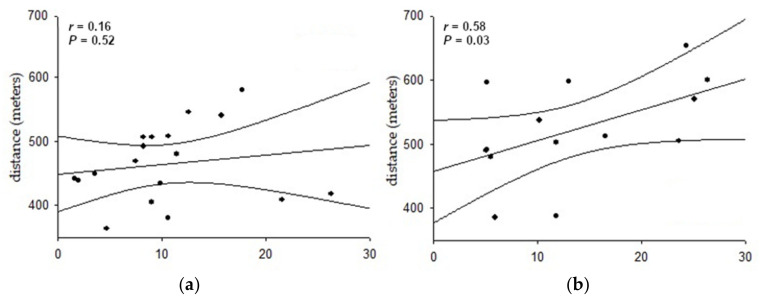
Correlation between the distance covered in the 6 min walk test and the 2LV% index of heart rate variability acquired in the supine position from individuals with systemic arterial hypertension (n = 32). (**a**) Eighteen post-COVID-19 individuals—G1, 2LV% from post-COVID-19 individuals; (**b**) fourteen non-COVID-19-diagnosed—G2, 2LV% from non-COVID-19-diagnosed.

**Table 1 jpm-13-01391-t001:** Characteristics of the individuals with systemic arterial hypertension (SAH) divided into infected with SARS-CoV-2 (SAH + COVID-19) or without clinical diagnosis of COVID-19 (SAH).

Variables	Total (N = 40)	SAH + COVID-19 G1 (n = 21)	SAH G2 (n = 19)	*p*
Age ^1^, years	53 ± 12	53 ± 13	53 ± 11	1.000
Male gender, n (%)	22 (55%)	9 (43%)	13 (68%)	0.105
Height ^1^, meters	1.65 ± 0.11	1.63 ± 0.10	1.67 ± 0.12	0.258
Total body mass ^1^, kg	85.8 ± 18.4	87.7 ± 22.4	83.9 ± 13.9	0.528
Body mass index ^1^, kg/m^2^	31.4 ± 5.4	32.6 ± 6.8	29.9 ± 3.4	0.126
Antihypertensives use, n (%)	40 (100%)	21 (100%)	19 (100%)	1.000
Obesity, n (%)	21 (53%)	12 (57%)	9 (47%)	0.382
Sedentary lifestyle, n (%)	25 (63%)	15 (71%)	10 (53%)	0.220
Smoking, n (%)	2 (5%)	0 (-)	2 (11%)	0.127
Hypothyroidism, n (%)	2 (5%)	2 (10%)	0 (-)	0.168
Dyslipidemia, n (%)	11 (28%)	6 (29%)	5 (26%)	0.873
Statins use, n (%)	7 (18%)	4 (19%)	3 (16%)	0.073
Diabetes mellitus, n (%)	11 (28%)	6 (29%)	5 (26%)	0.873
Hypoglycemics use, n (%)	11 (28%)	6 (29%)	5 (26%)	0.873

^1^ Data are shown as mean ± standard deviation.

**Table 2 jpm-13-01391-t002:** Comparison of linear and non-linear analysis of heart rate variability (HRV) indices in individuals with systemic arterial hypertension stratified into SARS-CoV-2-infected (G1) and non-COVID-19-diagnosed (G2) groups (n = 40).

HRV Indices	G1 (n = 21) Mean ± SD	G2 (n = 19) Mean ± SD	*p*	*d*
Linear analysis				
Mean iRR, ms	784.9 ± 109.3	848.0 ± 169.9	0.291	0.44
iRR variance, ms^2^	540.9 ± 484.5	787.8 ± 761.1	0.343	0.39
Spectral analysis				
LFun, un	44.1 ± 21.0	45.7 ± 21.1	0.811	0.07
HF, ms	165.6 ± 192.3	298.3 ± 489.4	0.725	0.36
Non-linear analysis				
0V%	22.1 ± 14.7	20.2 ± 12.8	0.168	0.14
1V%	46.4 ± 6.9	44.0 ± 4.8	0.873	0.41
2LV%	10.7 ± 6.9	13.4 ± 7.8	0.073	0.36
2UV%	20.8 ± 11.9	22.4 ± 13.5	0.873	0.13
Entropy				
Shannon entropy	3.51 ± 0.44	3.62 ± 0.40	0.410	0.26
Complexity index	1.07 ± 0.18	1.10 ± 0.19	0.660	0.16
Normalized complexity index	0.76 ± 0.09	0.74 ± 0.09	0.542	0.22

SD: standard deviation; iRR: R–R interval; LFun: low frequency expressed in normalized units; HF: high frequency; 0V%: patterns without variations; 1V%: patterns with one variation; 2LV%: patterns with two equal variations; 2UV%: patterns with two different variations.

**Table 3 jpm-13-01391-t003:** Comparison of the functional capacity by individuals with systemic arterial hypertension with low cardiorespiratory fitness stratified into SARS-CoV-2-infected or post-COVID-19 individuals (G1) and non-COVID-19-diagnosed (G2) groups submitted to 6 min walk test (n = 32).

Variables	G1 (n = 18)		G2 (n = 14)		MD	*p*	*d*
	Baseline	After	Baseline	After			
	Mean ± SD	Mean ± SD	Mean ± SD	Mean ± SD			
Heart rate frequency, bpm	79 ± 11	106 ± 16	74 ± 16	102 ± 25	4	0.580	0.19
Systolic blood pressure, mmHg	135 ± 12	137 ± 13	129 ± 25	132 ± 27	5	0.600	0.24
Diastolic blood pressure, mmHg	90 ± 9	90 ± 10	87 ± 18	87 ± 19	3	0.610	0.20
Perceived exertion (Borg scale), n	0 ± 1	3 ± 2	0 ± 1	3 ± 2	0	0.850	0.00
Oxygen saturation (SpO2), %	98 ± 1	97 ± 1	95 ± 1	95 ± 4	2	0.333	0.68
6 min walk distance covered, m	-	464.7 ± 59.5	-	522.2 ± 77.6	−58	0.024	0.83
Δ predicted and covered, %	-	85 ± 12	-	93 ± 10	−8	0.049	-

SD: standard deviation; MD: mean difference after the test between the two groups.

## Data Availability

Data from the present study can be made available for sharing purposes from the principal investigator upon reasonable request.

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
