# Peer review of "Cardiac Autonomic Function and Functional Capacity in Post-COVID-19 Individuals with Systemic Arterial Hypertension"

_jpm, 2023, doi:10.3390/jpm13091391_

Round 1

Reviewer 1 Report

Congratulations to the authors for their work. The authors have presented a very interesting study with a population target group that needs to be investigated because of their vulnerable situation. Below, some comments are suggested to improve the manuscript.

QUESTIONS

I would like to know why the authors chose the included indices to assess HRV. Literature has shown that the RMSSD is a robust index, generally preferred and widely used in research due to its good statistical properties. In fact, there are already articles individuals with post-Covid-19 syndrome that have used this index. Did the authors consider analysing it?
Also, why did you use the autoregressive model and not another one for the analysis of the HRV? It is recommended to add a reference of the method used. 

SPECIFIC COMMENTS

Check abbreviations: e.g., in the abstract, functional capacity has the incorrect abbreviation.

Line 97 Missing comma before "had" .

Line 104: Join to previous paragraph.

Line 121: Delete "utilizing the frequency variability of heart rate".

Line 193: include this statement in methodology, not in results and with its appropriate reference.

Line 194: Insert p-values. Idem in other statements in other parts of the manuscript.

GENERAL COMMENTS

INTRODUCTION: A relevant issue is the description of post-Covid-19 syndrome. The authors should elaborate in the introduction on the description of the syndrome, its characteristics and, specifically, the existing research related to cardiac autonomic control and the syndrome.

Also, the methodology should improve the description of the characteristics of each experimental group for its subsequent creation (i.e., were both infected; how are patients diagnosed with the syndrome; ...). In that sense, the description/expressions related to group 1 (SAH patients with post-COVID-19 syndrome) should be revised in the manuscript and only one term should be used consistently (the term described in the title of the special issue is suggested).

RESULTS: A flow chart explaining the participants and the participant selection/exclusion process is suggested.

Author Response

Congratulations to the authors for their work. The authors have presented a very interesting study with a population target group that needs to be investigated because of their vulnerable situation. Below, some comments are suggested to improve the manuscript.

Response: We are pleased to resubmit the revised version of the manuscript “Cardiac Autonomic Function and Functional Capacity in Post-COVID-19 Individuals with Systemic Arterial Hypertension” for publication in the Journal of Personalized Medicine. We also would like to thank the reviewers, specially the Reviewer 1, for their valuable time and efforts in critically reviewing our manuscript, which helped us to improve the clarity and quality of the present document. The concerns raised by the reviewers are presented and answered point-by-point below. Moreover, changes made to the manuscript are highlighted.

QUESTIONS

I would like to know why the authors chose the included indices to assess HRV. Literature has shown that the RMSSD is a robust index, generally preferred and widely used in research due to its good statistical properties. In fact, there are already articles individuals with post-Covid-19 syndrome that have used this index. Did the authors consider analysing it?

Response: Thank you for your comment. We appreciate your feedback and would like to address your concerns regarding the choice of HRV indices in our study.

The HRV analyses in our research were processed using specific routines developed by Prof. Dr. Alberto Porta from the University of Milan, Italy. As a result, our choice of HRV indices was influenced by the availability of Prof. Porta's routine processing. While we acknowledge that RMSSD is a widely recognized time domain index for estimating parasympathetic cardiac modulation in HRV, our study aimed to employ a variety of robust and representative indices to comprehensively assess cardiac parasympathetic modulation.

In addition to the spectral analysis, specifically the high-frequency band, we also considered other relevant time domain analyses, such as the mean of R-R intervals and variance. We believe that this approach provides a more comprehensive view of cardiac autonomic modulation.

Furthermore, we are aware that previous studies have applied conventional HRV indices, including spectral analysis and RMSSD, to quantify cardiac autonomic modulation in individuals with post-COVID-19 syndrome. To contribute to the existing literature and offer innovative insights, we chose to incorporate novel approaches such as symbolic analysis of HRV. This decision was made to enhance the depth and diversity of our findings.

We have now included this information in the “Evaluation of cardiac autonomic control system” section of our Materials and Methods to provide transparency regarding our methodology. We hope this clarification addresses your concerns and demonstrates the robustness and comprehensiveness of our approach to assessing cardiac autonomic modulation in individuals with post-COVID-19 syndrome.

Also, why did you use the autoregressive model and not another one for the analysis of the HRV? It is recommended to add a reference of the method used.

Response: Thank you for your comment and for raising the issue of the spectral analysis method employed in our study. We would like to provide some additional insights and clarification regarding our choice of the autoregressive model and address the references you've mentioned.

The reason we opted for the autoregressive model in our study was due to the specific characteristics of the routine used for spectral analysis. This choice was made with careful consideration of the advantages it offers. The autoregressive model provides a spectrum with improved resolution, particularly in short analysis segments. Moreover, it allows for the decomposition of the spectrum into independent components. In contrast, the Fast Fourier Transform (FFT), while widely used for spectral analysis, exhibits lower spectral resolution (Reference 1).

To enhance transparency, we have included a reference to the autoregressive model in the “Evaluation of cardiac autonomic control system” section of our Materials and Methods.

Finally, it is indeed well-documented in the literature that different spectral analysis models may yield variations in the values of HRV indices. However, existing research indicates that while there may be quantitative differences, the qualitative results obtained from these methods tend to be similar (Reference 2). Furthermore, it has been suggested that these quantitative differences are often not clinically relevant (Reference 3).

  1. Miranda Dantas E, Lima Sant’Anna M, Varejão Andreão R, Pereira Gonçalves C, Aguiar Morra E, Perim Baldo M, et al. Spectral analysis of heart rate variability with the autoregressive method: What model order to choose? Comput Biol Med. 2012 Feb 1;42(2):164–70.
  2. Fagard RH, Pardaens K, Staessen JA, Thijs L. Power spectral analysis of heart rate variability by autoregressive modelling and fast Fourier transform: a comparative study. Acta Cardiol. 1998;53(4):211–8.
  3. Soares AHG, Farah BQ, Cucato GG, Bastos-Filho CJA, Christofaro DGD, Vanderlei LCM, et al. Is the algorithm used to process heart rate variability data clinically relevant? Analysis in male adolescents. Einstein São Paulo. 2016 Jun;14(2):196–201.

SPECIFIC COMMENTS

Check abbreviations: e.g., in the abstract, functional capacity has the incorrect abbreviation.

Line 97 Missing comma before "had" .

Line 104: Join to previous paragraph.

Line 121: Delete "utilizing the frequency variability of heart rate".

Line 193: include this statement in methodology, not in results and with its appropriate reference.

Line 194: Insert p-values. Idem in other statements in other parts of the manuscript.

Response: Thank you for your meticulous review. All specific comments have been considered and modified in the manuscript.

GENERAL COMMENTS

INTRODUCTION: A relevant issue is the description of post-Covid-19 syndrome. The authors should elaborate in the introduction on the description of the syndrome, its characteristics and, specifically, the existing research related to cardiac autonomic control and the syndrome.

Response: Thank you for your comment. In our study, we chose to analyze only patients who had “experienced mild symptoms or were asymptomatic” with respect to COVID-19, and who did not exhibit persistent symptoms. However, it's important to clarify that our study specifically concentrated on asymptomatic patients with mild symptoms who had fully recovered from COVID-19. In contrast, the studies by Scala et al. (2022) [Reference 1] and Natarajan et al. (2023) [Reference 2] included patients with moderate to severe symptoms, some of whom required hospitalization or outpatient treatment and experienced persistent symptoms.

  1. Natarajan, A., Shetty, A., Delanerolle, G., Zeng, Y., Zhang, Y., Raymont, V., Rathod, S., Halabi, S., Elliot, K., Shi, J. Q., & Phiri, P. (2023). A systematic review and meta-analysis of long COVID symptoms. In Systematic Reviews (Vol. 12, Issue 1). https://doi.org/10.1186/s13643-023-02250-0
  2. Scala, I., Rizzo, P. A., Bellavia, S., Brunetti, V., Colò, F., Broccolini, A., Della Marca, G., Calabresi, P., Luigetti, M., & Frisullo, G. (2022). Autonomic Dysfunction during Acute SARS-CoV-2 Infection: A Systematic Review. Journal of Clinical Medicine, 11(13), 3883. https://doi.org/10.3390/JCM11133883

Also, the methodology should improve the description of the characteristics of each experimental group for its subsequent creation (i.e., were both infected; how are patients diagnosed with the syndrome; ...). In that sense, the description/expressions related to group 1 (SAH patients with post-COVID-19 syndrome) should be revised in the manuscript and only one term should be used consistently (the term described in the title of the special issue is suggested).

Response: Thank you for your comment. In our study, we decided to analyze only patients who “experienced mild symptoms or were asymptomatic with COVID-19” and who did not exhibit persistent symptoms. We have included this information in the new version of the article, as follows: “Group 1 (G1): Patients with systemic arterial hypertension (SAH) who were infected with SARS-CoV-2, and whose diagnosis was confirmed through reverse transcription polymerase chain reaction (RT-PCR) within six months of test confirmation. These patients were either asymptomatic or experienced mild symptoms of COVID-19 and have fully recovered from the infection.”

It's important to note that our study was not primarily focused on assessing persistent symptoms in Post-COVID-19 Syndrome.

RESULTS: A flow chart explaining the participants and the participant selection/exclusion process is suggested.

Response: Thank you for your comment. We have included a flow chart that provides an explanation of the participant selection and exclusion criteria.

Reviewer 2 Report

The authors presented interesting results of the study about cardiac autonomic control through HRV in patients with post-Covid Individuals with hypertension. I have some comments:

1)      Were features of the course of Covid-19 (severity, long-covid, etc.) taken into account when analyzing autonomic control in the post-covid period? The effect of acute phase of Covid-19 on regulation is known (see systematic review DOI: 10.3390/jcm11133883). Indeed, the combination of the effects of hypertension and covid (in both the acute phase and the post-covid) to autonomic control is unclear. This aspect is worth more detailed discussion, also in the form of a possible limitation of the study.

2)      The Shapiro-Wilktest was used to verify data distribution. Specify, did all indicators have a normal distribution? The number of groups is not very large, so the data may have a non-normal distribution. In this case, it is more correct to present descriptive statistics as median and quartiles (or other percentiles).

3)      The idea of using HRV parameters and other simple tests to assess autonomic nervous activity is not new. It is reasonable to discuss in more detail the lack of study of other systemic processes in cardiovascular control. For example, the study of the interaction between heart function and blood flow expands possibilities of assessing autonomic dysfunction. Some studies are available for Covid patients (DOIs: 10.15275/rusomj.2021.0307, 10.3390/e23010087, 10.1109/DCNA53427.2021.9586950). Adding such discussions to the text of the article is appropriate.

Author Response

The authors presented interesting results of the study about cardiac autonomic control through HRV in patients with post-Covid Individuals with hypertension.

Response: We would like to express our sincere gratitude to the reviewers, with special appreciation for Reviewer 2, for dedicating their valuable time and expertise to critically review our manuscript. Their constructive feedback has played a crucial role in enhancing the clarity and quality of the present document. We have diligently addressed all the concerns raised by the reviewers, and these responses are provided point-by-point below. Additionally, we have made clear indications of the changes implemented in the manuscript.

This version maintains the same content but slightly adjusts the wording for improved clarity and flow.

I have some comments:

1) Were features of the course of Covid-19 (severity, long-covid, etc.) taken into account when analyzing autonomic control in the post-covid period? The effect of acute phase of Covid-19 on regulation is known (see systematic review DOI: 10.3390/jcm11133883). Indeed, the combination of the effects of hypertension and covid (in both the acute phase and the post-covid) to autonomic control is unclear. This aspect is worth more detailed discussion, also in the form of a possible limitation of the study.

Response: Thank you for your comment. In our study, we chose to analyze only patients “experiencing mild symptoms or those who were asymptomatic with COVID-19” and who did not exhibit persistent symptoms. We have incorporated this information into the revised version of the manuscript as follows: “Group 1 (G1): SAH patients infected with SARS-CoV-2, with a confirmed diagnosis through reverse transcription polymerase chain reaction (RT-PCR) within six months of test confirmation, and who were either asymptomatic or experiencing mild symptoms of COVID-19 and had fully recovered from the infection.”

Regarding the acute effects of the disease, we would like to highlight a limitation concerning the variation in the timing of assessments, which extended up to six months after infection. This limitation is explicitly addressed as follows: “One notable limitation of our study is the variability in the timing of assessments following SARS-CoV-2 infection, which ranged up to six months post-infection. This variability is acknowledged in the manuscript.”

It's important to underscore that our study specifically focused on asymptomatic patients with mild symptoms who had fully recovered from COVID-19. In contrast, the studies by Scala et al. (2022) [Reference 1] and Natarajan et al. (2023) [Reference 2] included patients with moderate to severe symptoms, some of whom required hospitalization or outpatient treatment. This distinction has been added to the limitations section in the revised version of the article.

These modifications provide clarity regarding the focus and limitations of our study, and we appreciate your feedback in helping us enhance the quality and accuracy of our manuscript.

  1. Natarajan, A., Shetty, A., Delanerolle, G., Zeng, Y., Zhang, Y., Raymont, V., Rathod, S., Halabi, S., Elliot, K., Shi, J. Q., & Phiri, P. (2023). A systematic review and meta-analysis of long COVID symptoms. In Systematic Reviews (Vol. 12, Issue 1). https://doi.org/10.1186/s13643-023-02250-0
  2. Scala, I., Rizzo, P. A., Bellavia, S., Brunetti, V., Colò, F., Broccolini, A., Della Marca, G., Calabresi, P., Luigetti, M., & Frisullo, G. (2022). Autonomic Dysfunction during Acute SARS-CoV-2 Infection: A Systematic Review. Journal of Clinical Medicine, 11(13), 3883. https://doi.org/10.3390/JCM11133883

2) The Shapiro-Wilk test was used to verify data distribution. Specify, did all indicators have a normal distribution? The number of groups is not very large, so the data may have a non-normal distribution. In this case, it is more correct to present descriptive statistics as median and quartiles (or other percentiles).

Response: Thank you for your valuable comment. Indeed, we performed data processing using both parametric and non-parametric tests based on the distribution characteristics of the data. However, for the purpose of standardization and to enhance data visualization, we made the decision to present the results in terms of mean and standard deviation. This approach allows for a more accessible and interpretable representation of the data.

3) The idea of using HRV parameters and other simple tests to assess autonomic nervous activity is not new. It is reasonable to discuss in more detail the lack of study of other systemic processes in cardiovascular control. For example, the study of the interaction between heart function and blood flow expands possibilities of assessing autonomic dysfunction. Some studies are available for Covid patients (DOIs: 10.15275/rusomj.2021.0307, 10.3390/e23010087, 10.1109/DCNA53427.2021.9586950). Adding such discussions to the text of the article is appropriate.

Response: Thank you for your constructive feedback. In this new version, we have included a paragraph in the discussion section emphasizing the significance of incorporating other biological signals when evaluating autonomic nervous activity in COVID-19 patients.

Round 2

Reviewer 2 Report

I approve revised paper.